# The Change Rate of the *Fbxl21* Gene and the Amino Acid Composition of Its Protein Correlate with the Species-Specific Lifespan in Placental Mammals

**DOI:** 10.3390/biology13100792

**Published:** 2024-10-02

**Authors:** Vassily A. Lyubetsky, Gregory A. Shilovsky, Jian-Rong Yang, Alexandr V. Seliverstov, Oleg A. Zverkov

**Affiliations:** 1Institute for Information Transmission Problems of the Russian Academy of Sciences (Kharkevich Institute), 127051 Moscow, Russia; lyubetsk@iitp.ru (V.A.L.); gregory_sh@list.ru (G.A.S.); slvstv@iitp.ru (A.V.S.); 2Department of Genetics and Biomedical Informatics, Zhongshan School of Medicine, Sun Yat-sen University, Guangzhou 510080, China; 3Key Laboratory of Tropical Disease Control, Sun Yat-sen University, Ministry of Education, Guangzhou 510080, China

**Keywords:** bioinformatics, mammals, circadian rhythms, longevity, rate of change, amino acid composition, pseudogenization, gene loss

## Abstract

**Simple Summary:**

The relationship between genomic characteristics and species traits is of paramount importance for biology. For genomic characteristics, we considered characteristics close to the rate of gene change in the process of evolution, including such extreme cases as pseudogenization and gene loss. This can be conceptualized as selection at the gene level. Amino acid substitutions in the positions that functionally determine protein domains were also considered. For species traits, we considered the maximal reported lifespan, the body weight of an adult animal, and the related longevity quotient. The first and second traits are also related, but only at the stochastic level; therefore, their behavior is a priori not similar. We proposed a novel technique that allows one to determine the relationship between any genomic characteristic and species traits. This technique is exemplified in the physiologically significant genes involved in regulating circadian rhythms, which change quite rapidly during evolution. Regardless of devising this technique, the study of the genes that are critical for circadian rhythms is of interest on its own. For instance, we thoroughly examined the paralogous genes *Fbxl21* and *Fbxl3*, which are involved in the regulation of circadian rhythms. We found out that the above-mentioned characteristic of the *Fbxl21* gene correlates with the maximal reported lifespan and body weight only in two superorders of placental mammals, Euarchontoglires (the clades Euarchonta, Lagomorpha, and Rodentia) and Afrotheria. On the contrary, such a correlation is not observed in other superorders, such as Laurasiatheria and Xenarthra. The presence or absence of the correlation is confirmed statistically with a very high accuracy. The rate of change in the *Fbxl21* gene indicates, for example, the peculiarity of its evolution in apes.

**Abstract:**

This article proposes a methodology for establishing a relationship between the change rate of a given gene (relative to a given taxon) together with the amino acid composition of the proteins encoded by this gene and the traits of the species containing this gene. The methodology is illustrated based on the mammalian genes responsible for regulating the circadian rhythms that underlie a number of human disorders, particularly those associated with aging. The methods used are statistical and bioinformatic ones. A systematic search for orthologues, pseudogenes, and gene losses was performed using our previously developed methods. It is demonstrated that the least conserved *Fbxl21* gene in the Euarchontoglires superorder exhibits a statistically significant connection of genomic characteristics (the median of *dN*/*dS* for a gene relative to all the other orthologous genes of a taxon, as well as the preference or avoidance of certain amino acids in its protein) with species-specific lifespan and body weight. In contrast, no such connection is observed for *Fbxl21* in the Laurasiatheria superorder. This study goes beyond the protein-coding genes, since the accumulation of amino acid substitutions in the course of evolution leads to pseudogenization and even gene loss, although the relationship between the genomic characteristics and the species traits is still preserved. The proposed methodology is illustrated using the examples of circadian rhythm genes and proteins in placental mammals, e.g., longevity is connected with the rate of *Fbxl21* gene change, pseudogenization or gene loss, and specific amino acid substitutions (e.g., asparagine at the 19th position of the CRY-binding domain) in the protein encoded by this gene.

## 1. Introduction

Biorhythms are crucial for adaptation and depend on many factors. Since circadian rhythm disruption typically occurs upon aging, maintaining the proper function of the circadian rhythm system, underlying biorhythm regulation, holds great promise for improving the quality of life and increasing lifespan [1]. Studying the changes in the proteins that regulate circadian rhythms induced by pharmaceuticals or other genes is one of the primary goals of antiaging medicine [2]. For instance, the chronobiotics targeted at the Cry1 cryptochrome protein influence circadian rhythms and lifespan [1,3,4].

Based on the previously developed algorithms and devised software [5,6,7,8], this article examines the relationship between the maximum lifespan of species together with the body weight of an adult animal and the accumulation of changes during evolution of the species-specific genes listed above and their proteins, containing the Cry-binding domain (CBD), which contribute to the stability of cryptochromes.

**Circadian clock proteins**. Cryptochrome proteins (Cry1, Cry2) and period proteins (Per1, Per2, Per3) are involved in the mammalian circadian clock regulation [9]. They interact with other proteins such as Fbxl3, Fbxl21, β-TrCP, Fbxw11, GSK3α, GSK3β, CK1δ (Csnk1d), and CK1ε (Csnk1e). The alternative names of the proteins commonly used in literature are given in brackets. The names of the respective genes are spelled in a traditional way: *Btrc* (or *Fbxw1*). The core circadian rhythm mechanism also involves the Clock:Bmal1 heterodimer [10]. Per and Cry proteins are translocated into the nucleus, forming stable complexes, which include Cry1 and Cry2, and suppress the transcription of *Per1* and *Per2* genes by interacting with the transcriptional activators Clock and Bmal1. This causes mRNA levels to alternately rise and fall, followed by oscillations of the Per1 and Per2 protein levels within a period of approximately 24 h. Per phosphorylation by the casein kinase 1 epsilon (CK1ε) prevents its premature accumulation in the cytoplasm, inducing Per cytoplasmic degradation if it is not bound to Cry protein. The Per protein is less stable unless it is bound to Cry. Therefore, Per and Cry levels are controlled by ubiquitin-dependent proteasome degradation.

It is assumed that Cry protein mediates the adjustment of the circadian clock to the external lighting regime by inhibiting *Per* transcription and therefore regulating its level [11,12].

GSK3α and GSK3β kinases modify the products of almost all main genes responsible for circadian rhythms (*Bmal1*, *Clock*, *Per*, *Cry*, *Rev-erbα*). Modifications of the positive regulators of biorhythms lead to their inactivation and degradation, whereas modification of the negative regulators induces their activation and translocation into the nucleus [13,14]. GSK3β kinase, involved in the proteasome degradation system, also mediates cell aging and the development of many pathological processes, including sleep disorders and potentially leading to various neurodegenerative diseases.

*Gsk3a*, *Gsk3b,* and other genes contributing to circadian rhythm regulation, such as *Clock*, *Cry1*, and *Cry2*, are conserved in mammals [14]. For instance, the GSK3β protein encoded by the *Gsk3b* gene is conservative even in invertebrates and is almost identical in humans and macaques. The *Gsk3a* gene has no orthologs in birds and some reptiles. In contrast, the *Clock*, *Cry1,* and *Cry2* genes have orthologs in most vertebrates.

Mutations in casein kinase 1 (CK1) alter circadian behavior in various placental mammals [15,16]. Inhibiting CK1 by pharmaceuticals can be used for therapy in aberrant circadian rhythms [2,17]. CK1δ (Csnk1d) deficiency extends the circadian period, in contrast to the deficiency of CK1ε (Csnk1e) [18].

The system of regulating the mammalian circadian clock is complex and remains enigmatic. A possible scheme is shown in Figure 1, and questions concerning this regulation are discussed, among others, in [9]. However, in the present work, we touch upon only one aspect of the system at the end of this section. Of note, the regulation of circadian rhythms has been studied not only in mammals but also, for example, in *Culex pipiens* mosquitoes [19] and in *Drosophila* populations having different lifespans [20]. Meanwhile, these studies were performed within biological frameworks and did not address the question of statistically processing the relationship between species and genomic characteristics.

**Circadian rhythm protein degradation, the Fbxl3 protein.** The ubiquitin–proteasome protein degradation system is essential for various biological processes and is mediated by three enzymes working together: the ubiquitin-activating enzyme E1, the ubiquitin-conjugating enzyme E2, and the ubiquitin ligase E3 [21,22,23,24]. The latter makes ubiquitination specific by identifying the target substrates and mediating the transfer of ubiquitin from the ubiquitin-conjugating enzyme E2 to the substrate. The most well-known subfamily of the cullin-type E3 enzymes includes the Skp1-Cul1-F-box (SCF) complex [25]. It has four subunits: Skp1, Cul1, Rbx1, and the F-box protein with the characteristic spatial structure.

The Fbxl3 protein (like Fbxl21 below) is an F-box protein responsible for the ubiquitination and subsequent degradation of Cry cryptochromes, thus regulating the circadian rhythms. The formation of the SCF-Fbxl3 complex is regulated by cryptochrome binding [26]. The domain repositioning analysis showed that the C-terminal leucine-rich-repeat domain of Fbxl3 weakens the interaction with Skp1. This implies that a currently unknown protein interacts with the Fbxl3 C-terminal domain of the SCF-Fbxl3 complex formation.

To summarize, the SCF-Fbxl3 complex is responsible for the ubiquitin-dependent degradation of the Cry1 and Cry2 proteins, while Fbxl3 mutations extend the circadian clock cycle in mice [27,28,29]. Therefore, Fbxl3 is crucial for modulating Cry degradation during the circadian cycle.

**Paralogous proteins β-TrCP and Fbxw11** (also known as β-TrCP1 and β-TrCP2) are targeted at Per and also cause its degradation, which depends on Per phosphorylation by casein kinases CK1δ (Csnk1d) and CK1ε (Csnk1e), [26].

**The Fbxl21 protein (F-box and leucine-rich repeat protein 21)** is closely related to circadian rhythm regulation [30,31,32]. Moreover, the expression of *Fbxl21* is tissue-specific. The highest *Fbxl21* expression is observed in the suprachiasmatic nucleus (SCN) of the hypothalamus [30,33,34], the Master Clock responsible for the biorhythms that govern the processes of aging and longevity [35,36].

The genomes of some species, including humans and gorillas, contain the pseudogenes related to the *Fbxl21* gene [6,7]. For instance, in humans, this pseudogene is transcribed and may be related to severe diseases, including schizophrenia development [37,38]. The *Fbxl21* gene is a paralog of *Fbxl3*, and their evolutionary divergence was relatively recent: unlike *Fbxl3*, *Fbxl21* has not been identified in fish genomes. In Fbxl3 and Fbxl21, from *Mus musculus*, the identity of the amino acid sequences of the Cry-binding domain (CBD) is as high as 79%.

To summarize, Fbxl21, Fbxl3, β-TrCP, Fbxw11, GSK3α, GSK3β, CK1δ, and CK1ε interact with ubiquitin ligase adaptors, thus promoting the targeted proteasomal degradation. The concentration of the target proteins oscillates, establishing a circadian rhythm. Circadian clock disruption impairs sleep architecture and cognitive functions and can promote the development of neuropsychiatric disorders [14]. The GSK-3β-FBXL21 functional axis controls Telethonin (also known as Titin cap or TCAP) degradation by forming the SCF complex and thus regulates skeletal muscle functions. The Fbxl21 gene interacts with TCAP in a circadian manner, preventing its accumulation in skeletal muscle, while GSK3β phosphorylates Fbxl21 and TCAP, facilitating the Fbxl21-mediated, phosphodegron-dependent TCAP degradation. GSK3β inhibition or knockdown reduces the formation of the Fbxl21-Cul1 complex and hampers Fbxl21-mediated TCAP degradation. Fbxl21 hypomorphic mutation in Psttm mice disrupts circadian TCAP fluctuations. Psttm mice have substantial skeletal muscle defects, including abnormal fiber size, impaired exercise tolerance, reduced grip strength, and a response to glucocorticoid-induced muscle atrophy [39].

Although, in Laurasiatheria, we observe a considerable accumulation of the alterations in the *Fbxl21* gene, its protein product is also related to the species-specific lifespan (MRLS). For instance, the search for the genes related to longevity revealed four single-nucleotide polymorphisms associated with the extended lifespan of the Cane Corso dogs [40]. One of them, located in the third exon of *Fbxl21,* led to the substitution of arginine for tryptophan. This allele is associated with longevity in both homozygous and heterozygous forms.

**Fbxl21 and Fbxl3 comparison.** Although the amino acid sequences of mouse proteins Fbxl21 and Fbxl3 exhibit high similarity, they play nonidentical roles in circadian rhythm regulation [32]. Fbxl21 performs a dual function: it protects Cry from Fbxl3-mediated degradation in the nucleus and, at the same time, it promotes Cry degradation in the cytoplasm, ensuring the balance and cell compartmentalization of E3 ligases that compete for Cry-binding and fine-tuning the circadian period in mammals [41].

Along with Fbxl21, there are other mechanisms underlying circadian regulation. For instance, *TARDBP* encodes the TAR DNA-binding protein 43 (TDP-43), which stabilizes the Cry cryptochrome and suppresses Fbxl3 activity [32]. It is present in humans (GeneID 23435) and most mammals, which explains the existence of many species lacking *Fbxl21*. However, it is still unclear if *TARDBP* can fully compensate for the absence of *Fbxl21*.

This article proposes a methodology that establishes a relationship between the rate of change in a given gene (relative to a given taxon) together with the amino acid composition of its protein and the traits of the species harboring the gene. The study goes beyond the protein-coding genes, since the accumulation of amino acid substitutions during evolution may lead to pseudogenization or even gene loss, although the relationship between genomic characteristics and the species-specific traits remains preserved. The methodology is illustrated by the mammalian genes involved in the regulation of circadian rhythms, which contribute to a number of human disorders, particularly those associated with aging. The approach is applied to the following mammalian genes: *Btrc*, *Fbxw11* (*Btrc2*), *Fbxl21*, *Fbxl3*, *Gsk3a*, *Gsk3b*, *Csnk1d*, *Csnk1e*.

Therefore, in this paper, we examine the evolution of a panel of the proteins responsible for circadian rhythm regulation in placental mammals. Specifically, we established the association of the accumulated substitutions in the sequences of these proteins and the genes encoding them, including the related evolutionary events with the species-specific lifespan, body weight, and longevity quotient (LQ). In particular, here, we test the hypothesis that the accumulation of substitutions and even pseudogenization and loss of the *Fbxl21* gene are selectively associated (or not) in terms of MRLS and body mass in large taxonomic groups (level of superorders). In many species with high MRLS and body mass values, including humans, gorillas, and elephants, a significant change in this gene is also associated with a significant change in closely related species. In other words, this gene in Euarchontoglires and Afrotheria is under positive selection pressure, and in Laurasiatheria and Xenarthra, it is under stabilizing selection. Also, the second hypothesis runs as follows: the role of a single amino acid substitution in the Cry-binding domain of Fbxl21 is correlated with MRLS and body mass.

## 2. Materials and Methods

The data on the species-specific maximal reported lifespan (MRLS), body weight, longevity quotient (LQ), and reproductive performance were obtained from the AnAge database [42]; for *Galeopterus variegatus,* the estimated MRLS was 17.5 years [43]. Certain Chiroptera have high LQ and MRLS, as well as low body weight, which is particularly typical of *Myotis* (microbats) [44]. This characteristic is a distinctive feature of the order.

We studied 158 species of placental mammals from 18 orders. They comprise the existing representatives of Laurasiatheria, which include the following orders: Eulipotyphla, Chiroptera, Perissodactyla, Cetartiodactyla, Pholidota, and Carnivora. We also examined Euarchontoglires, which include the following orders: Lagomorpha, Rodentia, Scadentia, Dermoptera, and Primates. The Afrotheria superorder includes the following orders: Tubulidentata, Macroscelidea, Afrosoricida, Hyracoidea, Proboscidea, and Sirenia. The following number of species was considered in the above-mentioned orders: 1 species from Afrosoricida, 28 species from Artiodactyla, 36 species from Carnivora, 17 species from Chiroptera, 1 species from Cingulata, 1 species from Dermoptera, 3 species from Eulipotyphla, 2 species from Lagomorpha, 1 species from Macroscelidea, 4 species from Perissodactyla, 1 species from Pholidota, 1 species from Pilosa, 30 species from Primates, 1 species from Proboscidea, 28 species from Rodentia, 1 species from Scandentia, 1 species from Sirenia, and 1 species from Tubulidentata. The species were selected based on the quality of genome assembly and the availability of robust data on MRLS and body weight, which are unknown even for a number of species with a well-annotated genome. The sequences of proteins and transcripts have been obtained from the RefSeq database [45].

The search for orthologs, pseudogenes, and the genes lost during evolution was performed using the original methods described in [5,6,7,8]. These methods are based on homology and local synteny, along with computational parameters predicting orthology; the above-mentioned publications discuss the advantages (in our opinion) of these methods and the software used.

For each gene, we performed multiple sequence alignment (MSA) of the nucleotide sequences that encode the proteins under scrutiny, taking into account the codons in the open reading frames. We removed short sequences, low-quality proteins, and columns with deletions. Using the method and the software from the PAML package [46,47,48], we calculated the number of synonymous nucleotide substitutions *dS* (per synonymous site) and the number of nonsynonymous nucleotide substitutions *dN* (per nonsynonymous site). For each gene and all its orthologues in the specific taxonomic group, we calculated the *dN/dS* ratio median. This median can be considered an estimate of the *dN/dS* value for a particular gene and its common ancestor for all species in a taxonomic group. We interpret it as the relative rate of accumulation of substitutions and changes in a gene compared with this taxon up to the possible loss of a protein-coding gene.

We analyzed the nonannotated genome [49] of *Lepus timidus* by alignment of protein sequences with the in silico translated DNA according to the codons, using the Muscle method and the MEGA 7 software [50]. Scatter plots were prepared using the Seaborn library [51]. The protein tree was built using the Fast Minimum Evolution method and the COBALT web tool (the constraint-based multiple alignment tool).

## 3. Results and Discussion

### 3.1. The Relationship between MRLS (and Body Eeight) and the Substitution Accumulation Rate in the Genes Involved in Circadian Rhythm Regulation

In the introduction, we described the important function of the following mammalian genes: *Btrc*, *Fbxw11* (*Btrc2*), *Fbxl21*, *Fbxl3*, *Gsk3a*, *Gsk3b*, *Csnk1d*, *Csnk1e*. Particularly, they are involved in circadian rhythm regulation [1]. We examined a possible relationship between the substitution accumulation rate in these genes (including the formation of pseudogenes or even gene loss) and the MRLS and body weight. We used the median of the *dN/dS* values of a gene in relation to its orthologs in other species from the same mammalian taxon as the quantitative measure of the relative substitution accumulation rate in this gene. Further, we refer to it as “median (*dN*/*dS*)”, or simply “median”. Using the panel of the above-mentioned genes involved in circadian rhythm regulation, we developed an approach for studying the relationship between the characteristics of the genome and the species and further employed it to determine the relationship between the median and amino acid composition of a protein product with MRLS and body weight.

In the Euarchontoglires superorder which includes rodents and the Euarchonta grandorder, the variation interval of the *Fbxl3* gene median (depending on a gene) is much smaller than that of the *Fbxl21* gene: for *Fbxl3* and *Fbxl21*, it is [0.023, 0.054] and [0.074, 0.178], respectively (numbers in square brackets indicate the minimum and maximum of the median), which implies that the *Fbxl21* is far more sensitive to the changes in MRLS and typical adult body weight than its paralog *Fbxl3*, see Table 1, Table 2, Table 3 and Table 4 (the “median interval” column). Similarly, *Fbxw11*, the most conservative gene among the eight genes under scrutiny, as well as *Csnk1d*, has narrow variation intervals of the median: [0, 0.012] and [0.004, 0.032], respectively.

For the Euarchontoglires superorder, or for the unified Euarchontoglires and Afrotheria superorders, and even for placental mammals, we determined the incremental statistically significant dependencies between the *Fbxl21* median and MRLS (linear regression) as well as between the *Fbxl21* median and the typical adult body weight (exponential regression). Calculated *p*-values are: 10^−10^ in the first case (for MRLS) and 10^−7^ (for body weight); 8 × 10^−11^ (MRLS) and 7 × 10^−8^ (body weight) in the second case; 10^−6^ (MRLS) and 10^−8^ (body weight) in the third case (Table 1, Table 2, Table 3 and Table 4). It should be noted that 51 out of 61 Euarchontoglires species have the functional Fbxl21 protein, although we have no data on MRLS for *Microtus oregoni* and body weight for *Grammomys surdaster*.

We observed a unique/uncommon dependency for *Fbxw11.* In Euarchontoglires, its *p*-value for MRLS is 10^−3^ despite the lack of correlation in the case of body weight, whereas both correlations were absent in the unified superorders. However, in placental mammals, the *p*-value is 10^−6^ both for MRLS and body weight. No such relationships were observed for *Fbxl3*. We also confirmed the absence of these correlations in Laurasiatheria: the *p*-value was about 1 for all eight genes, except for *Fbxw11* and body weight, with a *p*-value equal to 7 × 10^−4^. These results are shown in Table 1, Table 2, Table 3 and Table 4; for *Fbxl21* and *Fbxl3*, data are also presented in Figure 2. Figure 2 also shows the uncertainty in the position of the regression line. The data used for calculations for *Fbxl21* and *Fbxl3* in Euarchontoglires and Laurasiatheria are shown in scatter plots in Figure 3 and Figure 4. In Euarchontoglires, the median correlates with the maximal reported lifespan and body weight for the *Fbxl21* gene, but such a correlation is absent for other genes. A correlation is also absent in Laurasiatheria.

We can conclude that for Fbxl21 the stabilizing selection prevails in Laurasiatheria whereas positive selection dominates in Euarchontoglires. Without any doubt, MRLS and body weight correlate in most mammal species.

Among Euarchontoglires that have the functional Fbxl21 protein, we revealed the statistically significant dependency between the relative substitution rate (the median) and MRLS (or body weight). On the contrary, we detected no dependency of the kind for the other examined genes, which are also involved in regulating circadian rhythms.

We would like to mention once again that Table 1, Table 2, Table 3 and Table 4 show the data and results for Euarchontoglires, for unified Euarchontoglires and Afrotheria, for Laurasiatheria, and for all placental mammals, respectively.

Figure 2 shows the obtained linear regressions for the median and MRLS (*A*), as well as for the median and the common logarithm of body weight (*B*) in the *Fbxl21* and *Fbxl3* genes from the Euarchontoglires superorder.

### 3.2. Pseudogenization and Fbxl21 Gene Loss

The relationship between the median and MRLS (or body weight) implies that in Euarchontoglires, with an increase in MRLS and body weight, as well as with changes in species-specific traits, important proteins (such as Fbxl21) accumulate mutations that can alter their function which may lead to pseudogenization and a subsequent gene loss.

The human pseudogene *FBXL21P* is a counterpart of the *Fbxl21* gene, which is confirmed by local synteny: conservative genes *IL9* and *LECT2* are closely located to the pseudogene [5,6,8]; *LECT2* is located on the complementary DNA strand and overlaps with *FBXL21P*. Similar findings are reported for the *IL9*, *FBXL21*, and *LECT2* genes in many other placental mammals.

In long-lived species, including humans, gorillas, and *Loxodonta africana* (Afrotheria, Proboscidea), *Fbxl21* became a pseudogene. Interestingly, in *Trichechus manatus* (Afrotheria, Sirenia) with MRLS of 69 years and body weight of 322 kg, the cysteine in the Cry-binding domain was substituted by tyrosine (C364Y mutation), resulting in substantial changes in protein properties. In this regard, the data on *Fbxl21* evolution in large Afrotheria (*Loxodonta africana* and *Trichechus manatus*) is in line with the results from human *Fbxl21*. It should be noted that in *Gorilla gorilla,* this pseudogene corresponds to the human pseudogene *FBXL21P*. In *Loxodonta Africana, Fbxl21* contains a termination codon in the Cry-binding domain at the consensus position corresponding to the leucine residue L370, similar to its flanking genes *IL9* and *LECT2*. Proboscideans have the longest MRLS of all Afrotheria, being third only to humans and baleen whales among mammals.

The human pseudogene *FBXL21P* has a matching gene (GeneID 101436012) in *Dasypus novemcinctus* (Xenarthra, Cingulata) with two noncoding LincRNA type transcripts (however, we found only the ENSDNOG00000052722 protein in the Ensembl database and a low-quality protein XP_023439683 in RefSeq, as well as the flanking gene *LECT2*). The *FBXL21P* pseudogene in *Oryctolagus cuniculus* has a matching gene (ENSOCUG00000030764) with two noncoding LincRNA-type transcripts. The same human pseudogene has a matching pseudogene with termination codons in *Lepus timidus*. The *Fbxl21* gene is absent in all known marsupials (the Marsupialia infraclass) and monotremes (the Monotremata order). It should be noted that the annotations in RefSeq sometimes include low-quality proteins such as Fbxl21p, in which the protein-coding region is terminated by a stop codon.

The lagomorphs do not follow the trend observed in Euarchontoglires: despite its MRLS and low body weight (the MRLS ranges between 7 and 18 years, and the body weight ranges between 100 g and 4.175 kg), lagomorph *Fbxl21* is a pseudogene. Typically, lagomorphs mostly follow the *r*-strategy [52,53], characterized by short life cycles and high fertility rates. However, there has been plentiful evidence of the animals being twice both the size and MRLS, such as rabbits [54]. In modern habitats, longer MRLS or higher body weight in lagomorphs is likely not adaptive and is not under positive selection pressure. Therefore, the deviation described above might be related to population factors.

The assumption formulated at the beginning of this section is in some ways true for anthropoid apes and the Cercopithecidae that represent a small part of the Euarchontoglires, which is interesting. Pseudogenization or, on the contrary, few substitutions in Fbxl21 are typical of most species in this group. Specifically, the *Fbxl21* gene underwent pseudogenization only in humans and gorillas (with long MRLS and high body weight). In contrast, other species belonging to the anthropoid apes and all Cercopithecidae have Fbxl21-like proteins containing a few substitutions (Figure 5). Although this finding seems to contradict our hypothesis, the Fbxl21 protein tree is not consistent with the phylogenetic tree, which divides the protein tree into clades with relatively high and low MRLS. The MRLS of chimpanzees is about 68 years (*Pan troglodytes* and *Pan paniscus* have identical proteins); *Pongo abelii*, which belongs to another clade, has an MRLS of 59 years. If we root the protein tree based on the root of the phylogenetic tree, we observe a high number of substitutions (distance) from the new root to *Pan* sp. (Figure 5). *Pan* sp. and, to a smaller extent, *Pongo*, represent outgroups in the phylogenetic tree; eliminating them from the protein tree makes it topologically close to the phylogenetic tree. In other words, in chimpanzees, the Fbxl21 protein differs considerably from the orthologous proteins in *Pongo abelii*, *Nomascus leucogenys*, *Hylobates moloch,* and all Cercopithecidae available for analysis (Figure 5). The common ancestor of chimpanzees has likely changed the direction of evolutionary changes, at least in the case of certain functions related to Fbxl21.

### 3.3. Amino Acid Substitutions in the Fbxl21 Protein

The difference in the occurrence of amino acids in the Fbxl21 protein between three superorders (Euarchontoglires, Laurasiatheria, and all placental mammals) is less than 0.5%. The amino acid occurrence is shown in Figure 6a.

In the Euarchontoglires superorder, MRLS exhibited a significant correlation with the occurrence of certain amino acids in the Fbxl21 protein sequence. For amino acids, the Pearson correlation coefficient *r* and *p*-value of the dependency are given in Figure 6b (for MRLS). In Euarchontoglires, we observed the preference (the positive *r*, MRLS rises with growing occurrence) of leucine (*p*-value is 10^−7^), glutamic acid (*p*-value is 10^−6^), and tyrosine (*p*-value is 10^−3^) and avoidance (the negative *r*, MRLS declines with growing occurrence) of valine (*p*-value is 10^−3^), arginine (*p*-value is10^−3^), and proline (*p*-value is 10^−9^). The most substantial preference and avoidance levels were observed for leucine and proline. This correlation does not depend on the amino acid occurrence, since we did not detect this correlation in the case of the amino acids with similar occurrence. The other amino acids in Euarchontoglires, as well as all amino acids in Laurasiatheria (the *p*-value is about 1), except histidine (with a *p*-value of 10^−4^), did not reveal any correlation of the kind. We obtained similar results (Figure 6c) for the amino acid occurrence and the body weight logarithm, which is in line with the correlation between MRLS and body weight. Therefore, Euarchontoglires exhibit a clear relationship between the genomic characteristics of the *Fbxl21* gene (its median), MRLS (and body weight), and the preference/avoidance of certain amino acids in the Fbxl21 protein. Low (close to zero) correlation coefficients *r* and high *p*-values observed for most amino acids may indicate that these amino acids have a lesser impact on MRLS, or they are highly conserved, preventing substitutions in the corresponding positions. Such positions in the Cry-binding domain are discussed below.

For leucine and proline, the *r* values (+0.62 and −0.7, respectively) can demonstrate the trend towards leucine accumulation and avoidance of proline with increasing MRLS. The relationship between MRLS and asparagine occurrence in the CBD domain, determining the function of *Fbxl21*, is discussed in the next section.

### 3.4. MRLS and Asparagine at the 19th Position of the Fbxl21 Cry-Binding Domain (CBD)

We obtained a consensus from the alignment of the Cry-binding domain (CBD), determining the function of *Fbxl21*. It is shown in the first row of Figure 7.

Only 12 positions of 54 positions in the domain fully match the consensus; and 12 and 8 more positions fully match the consensus in Euarchontoglires and Laurasiatheria, respectively. The latter positions almost never overlap with the former positions. Figure 7 shows the number of species from the taxon with residues that differ from the consensus, for each listed taxon. The last column shows the number of conservative positions in the consensus.

Analyzing the sequence alignment for placental mammals, we outlined the least conservative 19th position of the domain. In this position, asparagine is found in 56 species with a median LQ that is 17% higher than the median LQ of the remaining 90 species. The species composition in the former group indicates the relationship of this asparagine with MRLS and LQ. This group includes *Castor canadensis*, *Heterocephalus glaber*, *Nannospalax galili*, *Ictidomys tridecemlineatus*, *Marmota* spp., *Galeopterus variegatus*, Cercopithecidae, and anthropoid apes (except humans and gorillas that have *Fbxl21* as a pseudogene), as well as most cetaceans (except for *Physeter catodon*). The deviation for the lagomorphs was discussed above in the “Pseudogenization and *Fbxl21* Gene Loss” section.

The nature of the amino acid that substitutes another amino acid in a functionally important domain is of high relevance. We assigned six amino acid groups [55]. The first group contained cysteine; the second one, serine, threonine, alanine, glycine, and proline; the third one, aspartic acid, glutamic acid, asparagine, and glutamine; the fourth one, histidine, arginine, and lysine; the fifth one, methionine, isoleucine, leucine, and valine; the sixth one, tryptophan, tyrosine, and phenylalanine. We use the term “radical substitution” for the substitution that alters an initial amino acid for another one from the other group. The lower part of Figure 7 shows the number of radical substitutions for each taxon listed in this part. The 19th domain position is the least conservative in these taxa, since radical substitutions in this position occurred in 52 species. In these species, the median LQ exceeds the median LQ for the remaining 94 species of placental mammals by 35%. This indicates the correlation between MRLS and the 19th position of the Cry-binding domain of the Fbxl21 protein. Therefore, we can conclude that there is some relationship between MRLS and amino acid substitution in this domain and in Fbxl21 in general.

## 4. Conclusions

(1)The proposed method consists in determining the relationship between the species traits and a genomic characteristic, harnessing the methods of statistical analysis. The genomic characteristic is considered as the median *dN/dS* ratio.(2)For certain genes (such as *Fbxl21*), the accumulation of amino acid substitutions up to pseudogenization or gene loss, as well as the preference for certain amino acids in the encoded protein, is an effective way to achieve a significant phylogenetic change.(3)The *Fbxl21* gene and the species-specific maximal reported lifespan (MRLS), together with body weight, are examples of such a phylogenetic change in Euarchontoglires and Afrotheria, which is also observed in relatively small taxonomic groups, as, for example, in anthropoid apes and the Cercopithecidae.(4)In the Fbxl21 protein, changes were identified not only in the gene change rate for orders and superorders of mammals but also in the nature of the accumulated amino acid substitutions. For this protein, the stabilizing selection and the positive selection clearly prevail in Laurasiatheria (a sufficiently large *p*-value indicates the absence of correlation) and Euarchontoglires together with Afrotheria, where the *p*-value approximately equals 10^−10^, respectively.(5)In contrast, for proteins such as Fbxl3, β-TrCP, Fbxw11, GSK3α and GSK3β, CK1δ, and CK1ε, which are also closely related to circadian rhythm regulation in mammals, the stabilizing selection is characteristic of all mentioned superorders. The Bmal1, Clock, Pers, and Crys proteins, like many other circadian rhythm proteins, are highly conserved. For example, when cryptochrome circadian regulator 1 proteins were aligned in humans (586 aa) and mice (606 aa), 96% of the human protein and 93% of the mouse protein matched. The authors focused on studying circadian rhythm proteins, which change markedly during evolution.(6)We proposed a methodology to study the relationship between any genomic characteristics and species traits and illustrated its application on a number of circadian rhythm proteins. The authors hope that this methodology may be useful in other circadian rhythm proteins and within a different context.

## Figures and Tables

**Figure 1 biology-13-00792-f001:**
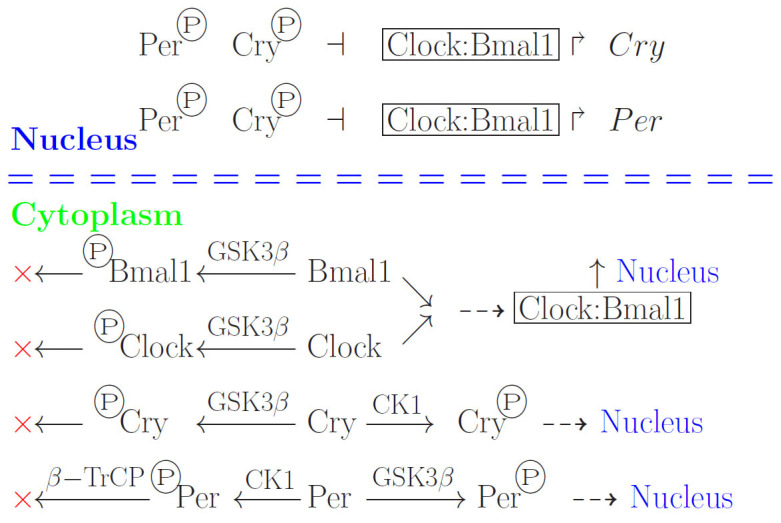
Molecular mechanism of circadian rhythms. The processes going on in the nucleus are shown at the top, and the processes in the cytoplasm are shown at the bottom. Dashed lines with arrows, translocation of the corresponding proteins into the nucleus; solid lines with arrows, direct effect, including catalysis; solid line with a blunt end, inhibition; circle with letter “P”, phosphate groups attached to the proteins; broken arrows, promoters; the diagonal crosses indicate protein degradation. Bmal1, basic helix-loop-helix ARNT like 1; CK1, casein kinase 1; Clock, circadian locomotor output cycles kaput protein; Cry, cryptochrome protein; GSK3, glycogen synthase kinase-3; Per, period protein; β-TrCP, beta-transducin repeat containing E3 ubiquitin–protein ligase.

**Figure 2 biology-13-00792-f002:**
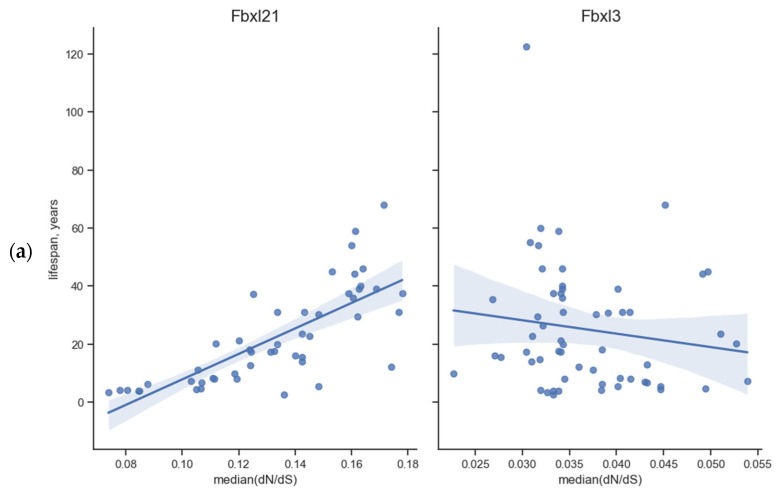
In the *Fbxl21* and *Fbxl3* genes from the Euarchontoglires superorder, for the species-specific maximal reported lifespan (MRLS) and the common logarithm of body weight, linear regressions (the lines with their estimated uncertainty) for the median and MRLS (**a**), as well as for the median and the common logarithm of body weight (**b**), are shown. Each dot represents an *Fbxl21* or *Fbxl3* gene from one animal species. The correlation between the considered species traits and the genomic characteristic—the median, which for *Fbxl3* does not seem reliable enough due to the ambiguity of the regression line and small interval of change in the median, is shown.

**Figure 3 biology-13-00792-f003:**
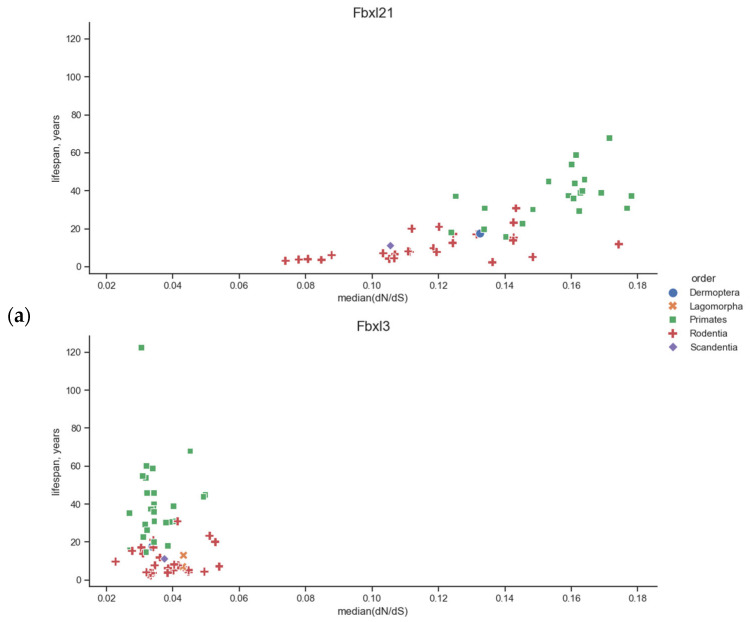
Scatter plots of the median and MRLS (**a**) and the common logarithm of body weight (**b**) for the *Fbxl21* and *Fbxl3* genes from the Euarchontoglires. Each dot represents an *Fbxl21* or *Fbxl3* gene from one animal species. The taxa are denoted by shape and color and are listed in the legend. The correlation between these species traits and the median is shown; see the description in Figure 2.

**Figure 4 biology-13-00792-f004:**
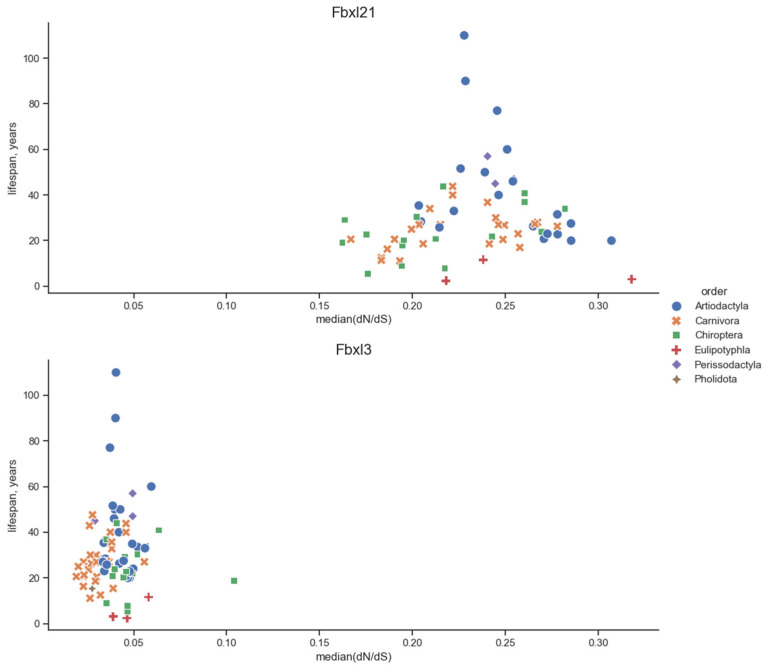
Scatter plots of the median and MRLS for the *Fbxl21* and *Fbxl3* genes in Cetartiodactyla (as a part of the Laurasiatheria superorder). Each dot represents an *Fbxl21* or *Fbxl3* gene from one animal species. The taxa are denoted with the shape and color and are listed in the legend. The correlation between the data on these species traits and the median is shown; see the description from Figure 2.

**Figure 5 biology-13-00792-f005:**
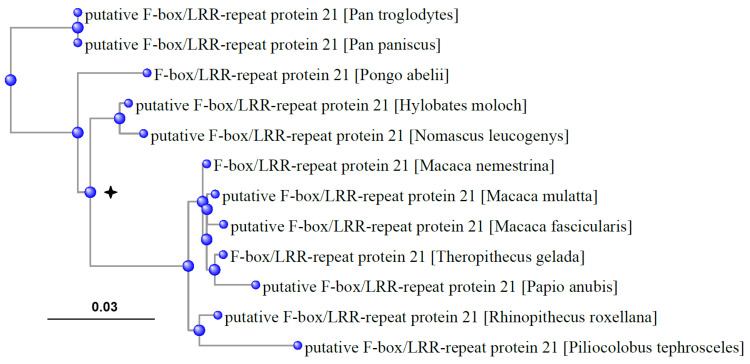
The protein tree of Fbxl21 in the anthropoid apes and Cercopithecidae (all species represented in RefSeq that have this protein). In the common chimpanzee and bonobo, the Fbxl21 protein sequences are identical. The node corresponding to the root of the phylogenetic tree is marked with a cross. The “0.03” above the bar represents the scale of the branch lengths, which corresponds to the expected number of substitutions per site. Specifically, a branch length of 0.03 indicates an expected 3% difference in amino acid sequence between the nodes connected by that branch. The peculiarity of *Fbxl21* gene evolution in apes is shown.

**Figure 6 biology-13-00792-f006:**
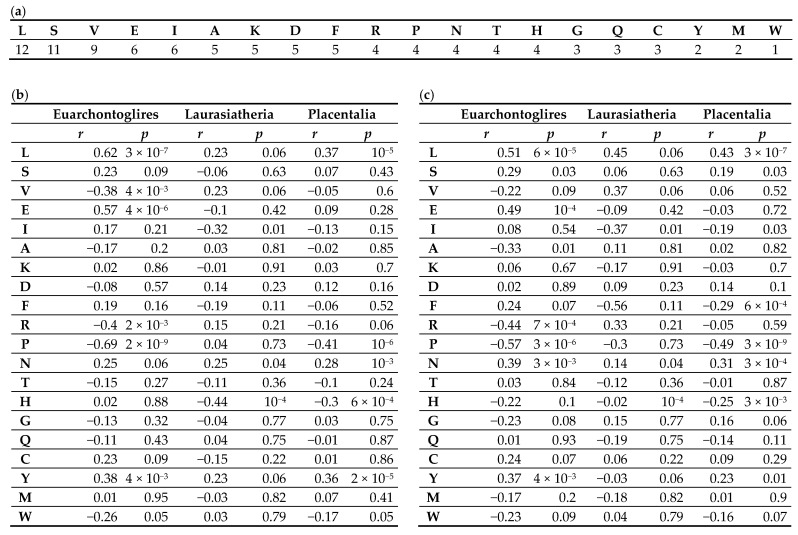
(**a**) The occurrence (%) of amino acids in the Fbxl21 protein, averaged by species. It is similar for Euarchontoglires, Laurasiatheria, and Placentalia. 20 amino acids are listed in the first row. (**b**) For the three superorders, Euarchontoglires, Laurasiatheria, and all placental mammals, the Pearson correlation coefficient *r* and the *p*-value of the amino acid occurrence in Fbxl21 with the species-specific lifespan are shown. (**c**) shows the common logarithm of the body weight; see (B) for a description. Correlation characteristics of the relationship between the frequency of each amino acid in the Fbxl21 protein in the indicated taxa and the species traits were obtained.

**Figure 7 biology-13-00792-f007:**
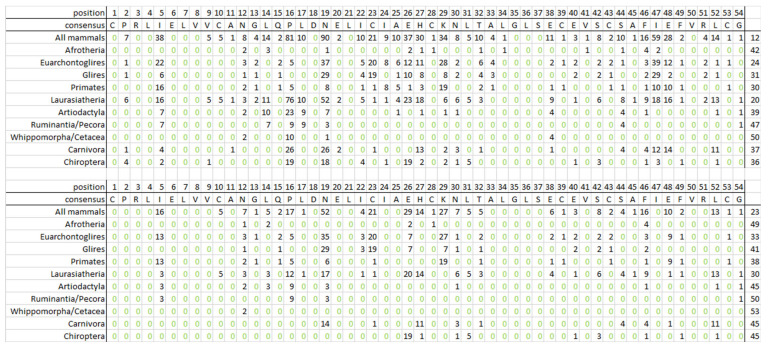
Alignment of Cry-binding domain. Almost all species in the list of mammal taxa include the Fbxl21 protein indicated in the rows. The table shows the consensus (54 positions) of the Fbxl21 Cry-binding domain and below the number of species in taxa with the amino acid distinct from the consensus amino acid. The upper part of the table shows all amino acid substitutions; the lower part shows only radical substitutions. The rightmost column shows the number of conservative positions where no substitution occurs, or no radical substitution in a taxon occurs, respectively. The 19th position of the domain is the least conservative.

**Table 1 biology-13-00792-t001:** In the *Fbxl21*, *Fbxl3*, *Fbxw11*, *Btrc*, *Gsk3a*, *Gsk3b, Csnk1d*, *Csnk1e* genes from the Euarchontoglires superorder, for the species-specific maximal reported lifespan (MRLS) and the common logarithm of the body weight, on the one hand, and the median, on the other hand, the following data are presented: the gene name; number *n* of the points used in linear regression; the linear correlation coefficient *r*; *p*-value (the probability of obtaining test results at least as extreme as the result actually observed, under the assumption that the null hypothesis is correct) assuming the lack of relationship between *y* and *x* (the null hypothesis); the linear regression equation *y = ax* + *b* (as the mean-square approximation of a given point set, Figure 2, Figure 3 and Figure 4), and the median variation interval (minimum and maximum values). Here, *y* indicates MRLS in years or the body weight logarithm in grams, while *x* indicates the median. Different scales of the coordinate axes explain the unusually high value of the coefficient of *x*. A highly significant correlation of the species characteristics, MRLS and body weight, with the proposed genomic characteristic—the median for the gene *Fbxl21* and a significant absence of such a correlation for other genes of the circadian rhythm are shown. The numerical characteristics of these correlations were also obtained.

Gene	*n*	*r*	*p*	Regression	Median Interval
median vs. lifespan					
*Fbxl21*	49	0.76	3 × 10^−10^	*y* = 439*x* − 36	[0.074, 0.178]
*Fbxl3*	58	−0.15	0.3	*y* = −463*x* + 42	[0.023, 0.054]
*Fbxw11*	54	−0.37	0.006	*y* = −1974*x* + 32	[0.000, 0.012]
*Btrc*	56	0.12	0.4	*y* = 34*x* + 23	[0.008, 0.265]
*Gsk3a*	52	−0.11	0.4	*y* = −135*x* + 30	[0.022, 0.147]
*Gsk3b*	52	−0.17	0.2	*y* = −102*x* + 28	[0.006, 0.225]
*Csnk1d*	56	0.14	0.3	*y* = 371*x* + 21	[0.004, 0.032]
*Csnk1e*	47	0.02	0.9	*y* = 10*x* + 25	[0.003, 0.136]
median vs. lg(weight)					
*Fbxl21*	49	0.67	10^−7^	*y* = 25*x* − 1	[0.074, 0.178]
*Fbxl3*	58	−0.17	0.2	*y* = −28*x* + 4	[0.023, 0.054]
*Fbxw11*	54	−0.35	0.01	*y* = −99*x* + 3	[0.000, 0.012]
*Btrc*	56	0.22	0.1	*y* = 3*x* + 3	[0.008, 0.265]
*Gsk3a*	52	−0.18	0.2	*y* = −12*x* + 3	[0.022, 0.147]
*Gsk3b*	52	−0.04	0.8	*y* = −*x* + 3	[0.006, 0.225]
*Csnk1d*	55	−0.04	0.8	*y* = −6*x* + 3	[0.004, 0.032]
*Csnk1e*	47	0.11	0.5	*y* = 3*x* + 3	[0.003, 0.136]

**Table 2 biology-13-00792-t002:** The data and results for unified Euarchontoglires and Afrotheria superorders; for designations, see the description for Table 1.

Gene	*n*	*r*	*p*	Regression
median vs. lifespan				
*Fbxl21*	53	0.75	8 × 10^−11^	*y* = 463*x* − 39
*Fbxl3*	63	−0.08	0.6	*y* = −249*x* + 35
*Fbxw11*	58	−0.35	0.007	*y* = −2016*x* + 33
*Btrc*	61	0.10	0.5	*y* = 31*x* + 25
*Gsk3a*	56	−0.05	0.7	*y* = −78*x* + 30
*Gsk3b*	57	−0.18	0.2	*y* = −115*x* + 29
*Csnk1d*	60	0.18	0.2	*y* = 483*x* + 20
*Csnk1e*	48	0.02	0.9	*y* = 15*x* + 25
median vs. lg (weight)				
*Fbxl21*	53	0.66	7 × 10^−8^	*y* = 27*x* − 1
*Fbxl3*	63	−0.08	0.5	*y* = −15*x* + 4
*Fbxw11*	58	−0.34	0.01	*y* = −111*x* + 3
*Btrc*	61	0.16	0.2	*y* = 3*x* + 3
*Gsk3a*	56	−0.10	0.5	*y* = −8*x* + 3
*Gsk3b*	57	−0.07	0.6	*y* = −2*x* + 3
*Csnk1d*	59	0.03	0.8	*y* = 5*x* + 3
*Csnk1e*	48	0.11	0.4	*y* = 4*x* + 3

**Table 3 biology-13-00792-t003:** The data and results for the Laurasiatheria superorder; for designations, see the description for Table 1.

Gene	*n*	*r*	*p*	Regression
median vs. lifespan				
*Fbxl21*	67	0.11	0.4	*y* = 58*x* + 16
*Fbxl3*	78	0.07	0.5	*y* = 105*x* + 27
*Fbxw11*	76	−0.29	0.01	*y* = −4552*x* + 32
*Btrc*	77	−0.08	0.5	*y* = −40*x* + 31
*Gsk3a*	75	0.33	0.004	*y* = 339*x* + 17
*Gsk3b*	77	0.38	0.0008	*y* = 79*x* + 26
*Csnk1d*	78	−0.10	0.4	*y* = −138*x* + 34
*Csnk1e*	42	0.29	0.06	*y* = 211*x* + 8
median vs. lg (weight)				
*Fbxl21*	67	0.26	0.03	*y* = 13*x* + 1
*Fbxl3*	78	−0.19	0.1	*y* = −26*x* + 5
*Fbxw11*	76	−0.38	0.0007	*y* = −686*x* + 4
*Btrc*	78	0.01	0.9	*y* = *x* + 4
*Gsk3a*	75	0.12	0.3	y = 12*x* + 4
*Gsk3b*	77	0.10	0.4	*y* = 2*x* + 4
*Csnk1d*	77	0.10	0.4	*y* = 14*x* + 4
*Csnk1e*	43	0.30	0.05	*y* = 24*x* + 2

**Table 4 biology-13-00792-t004:** The data and results for all placental mammals; for designations, see the description for Table 1.

Gene	*n*	*r*	*p*	Regression
median vs. lifespan				
*Fbxl21*	120	0.42	2 × 10^−6^	*y* = 166*x* − 4
*Fbxl3*	142	−0.00	1	*y* = −6*x* + 29
*Fbxw11*	135	−0.38	4 × 10^−6^	*y* = −2877*x* + 33
*Btrc*	140	0.01	0.9	*y* = 3*x* + 29
*Gsk3a*	132	0.22	0.01	*y* = 281*x* + 18
*Gsk3b*	136	0.22	0.01	*y* = 75*x* + 26
*Csnk1d*	140	0.01	0.9	*y* = 19*x* + 29
*Csnk1e*	91	0.02	0.8	*y* = 13*x* + 27
median vs. lg (weight)				
*Fbxl21*	120	0.48	4 × 10^−8^	*y* = 17*x* + 0
*Fbxl3*	142	−0.09	0.3	*y* = −16*x* + 4
*Fbxw11*	135	−0.40	2 × 10^−6^	*y* = −262*x* + 4
*Btrc*	141	−0.02	0.8	*y* = −*x* + 4
*Gsk3a*	132	0.14	0.1	*y* = 16*x* + 3
*Gsk3b*	136	0.15	0.08	*y* = 4*x* + 4
*Csnk1d*	138	0.11	0.2	*y* = 20*x* + 4
*Csnk1e*	92	0.12	0.3	*y* = 5*x* + 3

## Data Availability

The original contributions presented in the study are included in the article, further inquiries can be directed to the corresponding author.

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
