# Peer review of "The Change Rate of the Fbxl21 Gene and the Amino Acid Composition of Its Protein Correlate with the Species-Specific Lifespan in Placental Mammals"

_biology, 2024, doi:10.3390/biology13100792_

Round 1
Reviewer 1 Report
Comments and Suggestions for Authors
Circadian clocks are endogenous molecular machinery in cells, coordinating external environmental fluctuations to endogenous physiological and metabolic processes. Molecular mechanism driving circadian oscillations are highly conserved among organisms- feedback repressions formed by transcription factors and products of their transcribed genes, although primary sequences of circadian components may not be so. This paper analyzes the amino acid sequences of core circadian proteins in mammals and found that the Fbxl21 gene, encoding a ubiquitin ligase, correlates with organismal lifespan and body weight, while Fbxl21 does not. The paper is well written with appropriate citations and may be of interest to relevant researchers.
Comments:
The paper shows statistical correlations between residue variations and certain traits. Is there any experimental evidence to verify this finding?
The paper focuses on the primary amino acid sequences of core clock components. However, protein structures may not be altered despite residue changes. “19th position of the domain. In this position, asparagine is found in 56 species 427 with the median LQ, which is 17% higher than the median LQ” Does the mutation cause a structural change of the domain?
Tables 1-4. Does the analysis include Bmal, Clock, Pers, and Crys?
Figure 1. Each blue dot represents a Fbxl21 or Fbxl3 gene from an animal species?
Figures 2 and 3. The same question as for Figure 1: each symbol represents a species?
Figure 4. Where are Fbxl21 proteins from other species? What does “0.03” above the bar mean?
For all the figures, legends should contain detailed descriptions of the figures.
“by interacting with the transcriptional activators Clock and Bmal1. This causes mRNA levels to alternately rise and fall”
“[0.074, 0.178]”, 0.074 is the median while 0.178 is the interval, right?
Reviewer 2 Report
Comments and Suggestions for Authors
The authors use a molecular genetic approach to compare circadian rhythm genes/proteins in animals of the superorder Laurasiatheria (northern animals) and the superorder Euarchontoglires (southern animals), to which humans belong, and attempt to identify the relationship between the molecular evolution of circadian rhythm genes/proteins and longevity/body weight.
Introduction
Lines 38-47/171-176. The introduction as a whole has an inverted/broken unusual structure. It would not be a bad idea for the introduction to start with a general description of how the circadian rhythm system works and say where, in what organs it functions or refer to a contemporary review on the subject. Instead, in lines 38-47 the authors list what the paper is devoted to. Lines 38-47 should be moved to the end of the introduction and merged with lines 171-176.
Lines 60-74. It would have been helpful if the authors had drawn the mechanism of circadian proteins they describe.
Lines 97. The reference to the review about ubiquitin could be more up-to-date.
Lines 161-170. The article is about circadian rhythm genes/proteins, but in this part the authors talk a lot about encephalisation, brain development etc. Only the first sentence can be left out of this paragraph, the rest is not relevant to the topic of the article. Alternatively, the authors need to explain more clearly why they go on to encephalisation etc. when the article discusses circadian rhythm genes.
Lines 38-176. In general, the introduction needs to be rewritten and made to better reflect the article. For example, in its current form, the introduction does not tell why the authors compared animals from the northern and southern hemispheres and does not formulate a hypothesis as to how changes in circadian rhythm genes/proteins are related to longevity, although this is the focus of the paper, judging by the title.
Abstract
Overall, the abstract reflects an inverted pattern of introduction and it only talks about one gene, Fbxl21, although other genes have been investigated. The abstract needs to be modified.
Results and Discussion
Tables 1-4, although they contain the formal result of the work, can be moved to the supplimentary as the data they present are further used for plotting.
How might substitutions in the Cry-binding domain of the Fbxl21 protein affect the interaction with the Cry protein? Perhaps the authors could perform molecular dynamics analysis for selected pairs of proteins or perform an analogue of molecular docking for selected pairs of proteins to demonstrate that the substitutions they found could actually affect Cry/Fbxl21 interactions?
The paper lacks a separate ‘discussion’ section. Overall, there are very few references in the ‘results and discussion’ section and only lines 340-342 can be considered as a discussion. The results of the article are interesting, so the authors need to create a separate ‘discussion’ section and better discuss their own results by writing whether anyone else has obtained similar data both methodologically and factually. The authors' contribution to the problem of circadian gene variability is also not clear. What do the authors' results mean?
Conclusion
The authors should list under the figures the results they obtained.
Reviewer 3 Report
Comments and Suggestions for Authors
This article proposes a methodology for linking gene change rates and amino acid composition to species traits, using mammalian genes that regulate circadian rhythms. It shows that the Fbxl21 gene's genomic characteristics are significantly related to lifespan and body weight in the Euarchontoglires superorder but not in Laurasiatheria, highlighting the impact of amino acid substitutions and gene changes on longevity.
I only have a few questions and hope the authors could help answer them.
1. I understand the importance of studying biorhythms, but regarding the methodology, is there a specific reason why the author chose to focus on this subset of genes? Or did they randomly choose some physiologically significant genes?
2. Why did the author focus on the relationship between body weight and the studied genes? Is body weight a key factor in maximum reported lifespan (MRLS) or biorhythms?
3. Regarding the instructions, the author provided extensive background information on the genes of interest, which seems unrelated to the later analysis. However, it would be helpful if the author could emphasize the most important genes and present the information in a more logically organized manner.
Round 2
Reviewer 2 Report
Comments and Suggestions for Authors
My comments are highlighted in green. The article needs some more corrections.
